# The Origin of Capsid-Derived Immune Complexes and Their Impact on HBV-Induced Liver Diseases

**DOI:** 10.3390/v14122766

**Published:** 2022-12-12

**Authors:** Xiaonan Zhang, Yijie Tang, Min Wu, Cong Wang, Lyuyin Hu, Zhanqing Zhang

**Affiliations:** 1Centre for Research in Therapeutic Solutions, Biomedical Sciences, Faculty of Science and Technology, University of Canberra, Canberra, ACT 2617, Australia; 2Shanghai Public Health Clinical Center, Fudan University, Shanghai 201508, China

**Keywords:** HBV, chronic hepatitis B, capsid-antibody-complexes, liver fibrosis, liver inflammation

## Abstract

Over 240 million people worldwide are chronically infected with Hepatitis B Virus (HBV), a hepatotropic DNA virus with an evolutionary root of over 400 million years. Persistent HBV infection exhibits distinct and diverse phases of disease, from minimal liver pathology to fulminant Hepatitis, that vary in duration and severity among individuals. Although huge progress has been made in HBV research which has yielded an effective prophylactic vaccine and potent antiviral therapy, our understanding of its virology and immunobiology is still far from complete. For example, the recent re-discovery of serum HBV RNA in chronic Hepatitis B (CHB) patients has led to the proposal of noncanonical viral particles such as RNA virion and capsid-derived immune complex (Capsid-Antibody-Complexes, CACs) that contradict long-established basic theory. Furthermore, the existence of capsid-derived immune complex may hint at novel mechanism of HBV-induced liver disease. Here, we summarize the past and recent literature on HBV-induced immune complex. We propose that the release of capsid-derived particles by HBV has its deep evolutionary origin, and the associated complement activation serves as an indispensable trigger for intrahepatic damage and a catalyst for further cell-mediated immunopathology.

## 1. Introduction

Among the viruses that cause liver diseases, HBV is responsible for the greatest burden globally, with over 240 million people being chronic HBV surface antigen (HBsAg) carriers. HBV-induced chronic liver inflammation leads to liver fibrosis, cirrhosis, and hepatocellular carcinoma (HCC) [1]. HBV vaccination programs deployed in many countries have greatly lowered the HBV seropositive rate, particularly in the youngest age groups [2]. Nevertheless, due to unequal socioeconomic status, the decrease in viral prevalence is highly variable globally, with some areas still experiencing a rise. These facts highlight the difficulty in eradicating this disease solely by vaccination [3,4].

Being a member of the hepadnaviridae family with a partially double-stranded circular DNA genome, HBV infects humans and limited species of non-human primates [5]. The canonical model of HBV holds that the complete virion, also called the Dane particle, is composed of a viral genome covalently linked with viral polymerase, encapsidated by core antigens, and enveloped by an outer layer of viral surface antigens [6,7]. In addition, spherical or filamentous surface particles that are devoid of viral nucleic acid exist in a large quantity that dwarfs that of Dane particles. Once the persistent infection is established, a reservoir of covalently closed circular DNA (cccDNA) can be established as the genetic template for the viral replication cycle and constantly replenished by intracellular recycling of relaxed circular DNA. In the meantime, random integration of the linear viral genome is pervasive and contributes to the persistent antigenemia of HBsAg and carcinogenesis of HBV [8].

HBV-specific adaptive immune response, i.e., potent CD8 and CD4 T cell response and production of neutralizing antibody against HBsAg, was proven to be essential for acute resolution of the disease and prevention of re-infection as evidenced by chimpanzee studies [9,10]. Unfortunately, in chronic patients, HBV-specific CD8 T cells are depleted in number and are dysfunctional in target killing [11]. The humoral response in chronic disease is typified by a high titer of anti-HBcAg, however, this does not confer protection and even causes antibody-mediated liver damage [12]. The presence of antibodies against HBsAg is protective against further infection but is difficult to induce in CHB patients. The contemporary theory holds that as a non-cytopathic virus, the liver pathology caused by CHB is largely due to the inflammatory response that is primarily elicited by cell-based immunopathy executed by cytotoxic T lymphocytes and Natural Killer cells etc. [11].

## 2. New Surprises in HBV Study

Despite decades of active research, the canonical model of HBV virology and immunobiology as described above has recently encountered a series of challenges. In terms of viral particles, genome-free (empty) virions existing in large quantities in patient serum were identified [13]. More interestingly, a population of circulating HBV RNA has been rediscovered [14] that exists in quantities that are about 1/10 of HBV DNA {Butler, 2018 #151}. The factors that associate with the level of circulating HBV RNA are HBV genotype (higher in genotype B while lower in genotype D), basal core promoter mutation (lower HBV RNA) and ALT (positive correlation) [8,15]. The exact viral form that harbors HBV RNA is still under debate [16,17,18]. It is proposed that viral pregenomic RNA (pgRNA) is encapsidated and enveloped to form pgRNA virion [16]. Others reported HBV spliced variants as major species of HBV RNA [19]. By performing an unbiased analysis of cell culture supernatant and serum of CHB patients, we uncovered a much more complex picture of HBV-RNA-containing viral particles [20]. We put forward a hypothesis that the majority of the circulating RNA was enclosed in an alternative type, i.e., Capsid-Antibody-Complexes (CACs). This theory is based on several lines of evidence as follows. First, ultracentrifugation separation followed by northern blot analysis revealed that the bulk of extracellular HBV RNA is within the secreted viral capsid which is widely known to be actively released in hepatoma cell lines. The large majority of the RNA was heterogeneous in length as opposed to a uniform size distribution. Second, further investigation in serum of CHB patients showed that high level of HBV RNA correlated with the detection of immature viral DNA, mostly single-stranded DNA, which is known not to be released by enveloped virions [21]. Third, native agarose gel analysis demonstrated that these viral particles have a much slower electrophoretic migration pattern, distinct from that of the mature virions and naked capsid in patient serum. Lastly, the involvement of immunoglobulin is supported by the co-precipitation of viral RNA and DNA from serum of CHB patients using Protein A/G beads. The immature nature of these particles, the heterogeneous electrophoretic behavior and the participation of antibodies prompted us to hypothesize that HBV nucleocapsids with varying levels of maturity could be released into the circulation where they encounter a high titer of anti-core antibodies and form immune complexes, hence Capsid-Antibody-Complexes (CACs) [20]. The exact mechanism leading to the different egress pathways of virions and naked capsids is not well understood. As a major determinant in the HBV release process, the large, middle and small surface antigen is thought to be play a major role. Paradoxically, although the large surface antigen is essential for complete virion secretion [22], its overexpression caused intracellular retention of virions in the multivesicular bodies and heightened the release of naked capsid. Conversely, forced expression of M/S surface antigen promoted the secretion of virions and decreased the release of naked capsid [23]. These results suggest that the relative ratio of large, middle and small surface antigens modulate the molecular decision regarding the release of enveloped or unenveloped viral particles in a non-linear fashion.

## 3. Immune Complexes in Viral Infections

The hypothesis we proposed seemed to significantly deviate from the canonical theories. Nevertheless, immune complex formation during viral infection is not at all a new concept. Non-neutralizing antibodies induced after primary Dengue Virus infection can inadvertently enhance secondary infection due to the accelerated internalization of immune complex via Fcγ-receptor-mediated entry and cause serious complications of dengue hemorrhagic fever [24,25]. In Hepatitis C Virus (HCV) infection, viral particles and core proteins induce a subset of B cells to expand and synthesize a large amount of IgM with rheumatoid factor activity. These IgM molecules bind to HCV particles and form cold-precipitable, multimolecular immune complexes called cryoglobulins in 25 to 30% of individuals. Cryoglobulin-induced illness, known as cryoglobulinemic vasculitis, includes a variety of symptoms in the skin, kidney, musculoskeletal systems and nervous system [26].

Immune complexes were also reported very early on after the identification of HBV. In 1969, using a complement fixation assay, Shulman and Barker reported a high rate of “anticomplementary activity” in individuals infected with HBV [27]. Using electron microscopy, Almeida and Waterson further provided visual evidence of large aggregates in chronic active and fulminant Hepatitis [28]. They suggested the existence of the immune complex of “Australia antigen”, either in the condition of antigen excess or in the condition of antibody excess depending on the phase of the infection. One year after these reports, Gocke et al. reported significant prevalence (four in eleven cases) of Australia antigen in biopsy-proven polyarteritis nodosa. The deposition of the antigen on muscle tissues of the patient, as evidenced by immunofluorescence, supports the pathogenetic role of HBsAg-derived immune complex [29]. Similarly, Combes et al. reported the membranous glomerulonephritis associated with HBsAg antigenemia [30]. In a comprehensive retrospective study, Trepo et al. analyzed 55 cases with histologically confirmed polyarteritis and found that 69% of them had either HBsAg or anti-HBs. However, no correlation was found between immune complexes and liver disease suggesting that HBsAg immune complex was not pathogenic for the liver [31]. A more recent clinical study provided evidence that remission of polyarteritis nodosa is primarily related to control of HBV replication. No specific genetic variations in the HBV genome predispose to this disease [32].

## 4. Capsid-Derived Immune Complex: Revival of an Old Concept

While the existence of HBsAg-derived immune complex has been a consensus, the proposition of the widespread existence of capsid-derived immune complex in CHB patients has long been seen as unorthodox. Nevertheless, traces of evidence were also found in the literature. In 1979, De Vos et al. performed a comprehensive electron microscopy study on eighteen liver biopsies of HBsAg positive patients. In addition to the widespread HBsAg dots in the cisternae of the endoplasmic reticulum (ER) and naked capsid in the nucleus and nuclear pore, “core particles surrounded by a narrow cloud of semi electron dense material” were occasionally found in the space of Disse with appearance distinct from that of the Dane particle, which was observed as single core particle surrounded by a clear halo and a dark ring resided in the cisternae of the ER [33]. Similarly, Yamada et al. provided another electron microscopy observation. They found that in addition to the budding of core particles into the cisternae of ER, they were also often found adjacent to the cell membrane, suggestive of a direct budding from the surface of the cell [34]. In 1982, Trevisan et al. reported that after elution of the immunoglobulin from liver biopsy samples of HBV patients, the released IgG showed specificity against HBV core antigen instead of surface antigen [35]. Indeed, the enhancement of complement-mediated cytotoxicity after the addition of core-specific antibody was observed in ex vivo cultures of hepatocytes collected from CHB patients, and the magnitude of cytotoxicity seemed to be related to the severity of liver disease of the patients from who biopsies were taken [36].

Although the above observations hinted at the formation of CACs in vivo, the idea of direct release of core particles was still under debate at the time. Perhaps the most convincing evidence came from studies conducted by Moller et al. who investigated the longitudinal serological manifestations in HBV patients with concomitant immunodeficiency which resulted in a delayed production of antibody to core protein [37,38]. Remarkably, in a careful longitudinal observation of an acute Hepatitis B patient with acquired immunodeficiency syndrome (AIDS), they detected free, uncomplex HBcAg using radioimmunoassay and autoradiography and the appearance of free HBcAg preceded the elevation of serum ALT suggesting that it was not released by cell lysis [37]. Following up on this patient, using electron microscopy, they further demonstrated that the viral particles purified from patients with negative anti-HBc formed aggregates with exogenous monoclonal HBc antibody while the same procedure did not produce an aggregate in samples with positive anti-HBc [38]. These results unambiguously proved the release of core particles into circulation in a special condition of delayed anti-core production. In a common case of HBV infection, such events will be complicated by the rapid neutralization by anti-core antibody resulting in a heterogeneous population with varying molecular mass and in vivo kinetics of disposal.

It should be noted, however, that following the publications by Moller et al., Possehl et al. [39] conducted a similar investigation in HBV infected individuals who failed to develop anti-HBc. Using an ELISA assay specifically for core antigen (sensitivity 0.3 ng/mL), they did not detect free capsid in these individuals while stripping the surface antigen detected significant signal (as high as 20 ng/mL). Due to the very different patients and analytical methods used in the Moller and Possehl study, it is impractical to dismiss either side’s conclusion. It remains possible that the naked capsid consists of a low percentage (less than 1%), of the whole core-containing viral population in Possehl’s experimental condition. More recently, Hong et al. conducted a comprehensive analysis of the core and precore proteins in the supernatant of HBV-infected primary hepatocyte, HBV transgenic and humanized chimeric mice [40]. They detected a minute amount of capsid using ultracentrifugation followed by native gel agarose electrophoresis, which was interpreted as a result of cell lysis. Regardless of its origin, this nevertheless proved its existence, which may trigger a positive feedback amplification precipitated by CACs-induced inflammation in CHB as discussed below.

## 5. Passive or Active Release of HBV Capsid?

Ever since the concept of CACs formation in vivo was put forward, significant doubts over whether they are actively released or passively formed due to liver damage have been constantly raised. Although it is virtually impossible to dismiss the possibility of passive release just as one cannot completely dismiss the accidental release, via cell lysis, of a trace amount of Dane particle in a large population of virions, there are multiple pieces of evidence supporting the natural route of capsid release. Recent advancements in the evolutionary origin of HBV provided a unique perspective. By systematically analyzing the Sequence Read Archives deposited by numerous studies (>25,000 experiments) Lauber et al. recovered 17 complete genome sequences of HBV-related virus in teleost fishes. They found that 13 of the 17 genomes constitute a group dubbed “nackednavirueses” with all the key elements of hepadnaviruses except the preS/S ORF [41]. Revitalization of these genomes in cell culture reveals that they are replication-competent exogenous viruses with characteristic polymerase-primed replication initiation and are capable of producing non-enveloped extracellular progeny particles. Phylogenetic reconstruction of these genomes with all other HBV-related viruses indicates that the enveloped and non-enveloped viruses separated over 400 million years ago, before the rise of the tetrapod. A crucial implication of this study is that the release of naked capsids in human infection “might be a mere vestigial feature retained from their distant past as non-enveloped viruses” [41].

Despite the backing of the evolutionary clue, proving the active secretion of naked capsids in HBV-infected individuals is much trickier due to the intertwined nature of virological and immunological parameters in clinical observations leaving interpretation of causal relationship difficult. However, a longitudinal analysis of HBV infection in individuals with delayed anti-HBc development provided crucial insights [37]. Moller et al. followed the serological manifestations of a case of acute Hepatitis B with a transition to chronicity due to HIV co-infection. Strikingly, free HBcAg in serum was detected during the early acute phase prior to the rise of ALT which was followed by the delayed appearance of anti-HBc. This unequivocally shows that liver damage is not the prerequisite for the release of naked capsids. Nevertheless, it is plausible that the initial secretion of capsids, which form CACs in circulation, may lead to subsequent liver damage due to the intrinsic proinflammatory nature of immune complex. The CACs-prompted secondary inflammatory responses may further facilitate the passive release of naked capsids through a positive feedback loop mechanism known in autoimmune diseases, i.e., the early release of minute amount of autoantigens leads to chronic inflammation followed by further release of more antigens as a result of tissue damage [42]. If the level of serum HBV RNA were assumed to be a surrogate marker for CACs, then a significant correlation between HBV RNA and ALT level in a large group of 1409 untreated adults with chronic Hepatitis B [8] may be interpreted as supporting evidence. Furthermore, we have developed a novel assay directly quantifying serum CACs levels and preliminary data has shown its close association with serum ALT and intrahepatic inflammation (manuscript in preparation).

## 6. Possible Mechanism for CACs-Elicited Liver Pathology

Considering the intrinsic stimulatory nature of the immune complex, the notion that CACs may cause intrahepatic inflammation is hardly surprising. Nevertheless, the detailed mechanism leading to this outcome is largely undefined. The fragmented experimental evidence gathered so far may provide clues as follows (Figure 1).

### 6.1. Complement-Mediated Direct Lysis

As early as 1981, researchers had found that core specific IgG were bound to liver cell membranes in chronic Hepatitis [35]. Subsequent ex vivo studies showed that the addition of anti-HBc supplemented with complement in primary culture of hepatocytes isolated from chronic active patients resulted in enhanced cytotoxicity [36]. These data suggest that core particles may be present on the hepatocyte cell membrane and can bind to its antibodies leading to complement fixation and subsequent cell lysis. A more recent investigation revealed the humoral immune response against HBV core antigen may be responsible for the pathogenesis of HBV-induced acute liver failure. The authors suggest that when abundantly expressed, HBcAg is available on hepatocyte surface, and uninfected bystander hepatocytes may also bind to core antigen released from dying cells. They also demonstrated the potent lysis of hepatocytes by anti-core antibody and complement-mediated cell lysis [43].

### 6.2. Antibody-Dependent Cellular Cytotoxicity (ADCC)

Another feature of antibody-binding and complement engagement on target tissues is the accompanied cellular immune responses. NK cells armed with anti-core antibodies may execute the cell lysis. Indeed, hypercytolytic activity of NK cells correlates with liver injury in chronic Hepatitis B [44]. Whether the anti-core antibody is necessary for this activity remains to be tested.

### 6.3. Complement-Mediated Chemotaxis and Leukocyte Infiltration

Another possible mechanism is the recruitment of proinflammatory leukocytes by the by-products (e.g., C3a, C5a) of the complement activation cascade which exhibit chemotactic or vasoactive properties. It is worth noting that the role of portal myofibroblasts, which are mostly believed to be differentiated from hepatic stellate cells [45], in amplifying the leukocyte infiltration should not be overlooked. In particular, the complement activation and its subsequent deposition on myofibroblasts might serve as an important trigger for their activation which has been shown in a similar scenario in chronic kidney disease [46]. In addition to their ability to alter the extracellular matrix in the portal tract, activated myofibroblasts are highly chemotactic for lymphocytes which further promote their adhesion and invasion into the liver lobules [47].

### 6.4. Antigen Presentation Mediated by Core-Specific B Cells

An unexpected role of B cells and plasma cells in the pathogenesis of HBV-induced acute liver failure was revealed by a detailed histological, transcriptomic analysis of liver tissues [43,48]. Massive intrahepatic accumulation of cells expressing CD20 and IRF4 was documented in these tissues. The further sequencing of antibody variable regions demonstrated that the overwhelming B cell response was targeting the core antigen. It is conceivable that these cells are recruited to liver lobules via their surface B cell receptors having high affinity to HBV capsid and serving as intrahepatic antigen presentation cells [49] and promote subsequent antigen-specific CD4 and CD8 T cell response. Whether this mechanism also operates in immune-active chronic Hepatitis B remains to be demonstrated.

### 6.5. Epitope Spreading and the Nurturing of Autoimmunity

Finally, the accumulation of CACs within the liver lobule might also foster the production of autoantibodies such as anti-asialoglycoprotein receptors [50], which were repeatedly found in chronic active Hepatitis B [50,51,52]. Indeed, antibody to asialoglycoprotein receptor was induced after inoculation of woodchuck Hepatitis virus [53] and triggered complement-mediated hepatocytolysis [54]. A mechanism called “epitope spreading” whereby autoreactive CD4 T cells are activated within a proinflammatory intrahepatic milieu might encourage the production of autoantibodies by otherwise anergic B cells [42].

## 7. Conclusions and Future Directions

Despite its discovery for over half a century, the study of HBV continues to bring new surprises, from its evolutionary origin to its complex virological and immunological manifestations in vivo. The realization that hepadnaviruses originated from non-enveloped viruses reveals the de novo generation of the preS/S ORF through “overprinting” of the pre-existing polymerase-coding sequence in an alternative reading frame around 400 million years ago [41]. Although the exact evolutionary driver for this switch is unknown, acquisition of viral envelop may favor its specialization to a single organ (liver) and co-evolution with the host to evade the adaptive immune response [55,56] that is increasingly reactive toward viral capsid. The ancestral link with non-enveloped nackednaviruses also highlights the vestigial feature of active capsid release by HBV. This notion is also supported by virological and ultrastructural investigations in the clinic, particularly in cases with delayed anti-core production [37,38].

Although it is easy to see the immunogenic potential of capsid-derived immune complex, its exact role in the pathogenesis or resolution of HBV infection is largely unexplored. The lack of a sensitive and reliable test for CACs constitutes an immediate problem. We have developed a simple microwell-based assay which produces a semi-quantitative readout of the level of CACs in serum. A retrospective, cross-sectional study is underway to delineate its feature during the natural course of HBV infection. Preliminary data uncovered a close link between CACs levels and liver inflammation consistent with our expectation (manuscript in preparation). Further longitudinal studies on CHB patients undergoing antiviral therapy are also highly desirable to reveal the kinetics of CACs in relation to varying prognoses of treatment. Finally, the mechanism underlying CACs-mediated liver injury is vital for understanding the molecular and cellular players involved in this process. Histological and molecular pathology assays can provide spatially resolved, single-cell data which are crucial to decipher the cell types and molecular markers that are downstream of the CACs-initiated immunopathology. It is anticipated that the revitalization of the concept of capsid-specific immune complex and further exploration of its pathophysiological roles would lend unexpected insight into the pathogenesis of CHB and provide rationales for the development of next-generation therapeutics aiming at a functional cure.

## Figures and Tables

**Figure 1 viruses-14-02766-f001:**
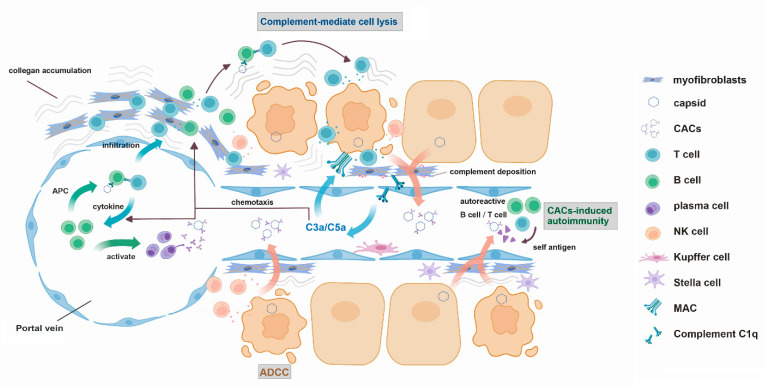
Proposed mechanism of CACs mediated liver injury. ADCC: antibody-dependent cellular cytotoxicity; MAC: membrane attack complex; APC: antigen presenting cell.

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
