# Peer review of "The Origin of Capsid-Derived Immune Complexes and Their Impact on HBV-Induced Liver Diseases"

_viruses, 2022, doi:10.3390/v14122766_

Round 1

Reviewer 1 Report

In their review entitled „The origin of capsid-derived immune complexes and their im-pact on HBV-induced liver diseases” Xionan Zhang and coworker summarize the literature on HBV-induced immune complexes. The authors hypothesize that the complement activation due to the release of capsid derived particles serves as trigger for intrahepatic damage.
The review is well written and clearly structured. The hypotheses raised in this review are interesting and worth to be discussed. There are just a few minor points:
-line 62: is there an impact of the HBV genotype on the mount of circulating RNA? What is the impact of HDV coinfection on this.
-which factors modulate the release of naked capsids, RNA loaded capsids and RNA harbouring virions
-line 171: secretion should be replace by release
-is there an evidence that CACs can be internalized by Fcgamme receptors in non-hepatic cells and cause subsequent expression of the packaged RNA.

Reviewer 2 Report

The manuscript Viruses-1915696 "The origin of capsid-derived immune complexes and their impact on HBV-induced liver diseases" authored by X. Zhang et al. attempts to elaborate on the hypotheses that the majority of the circulating in the blood HBV RNA was enclosed in an alternative type of vesicles (not HBV virions), i.e, HBV capsid-antibody-complexes (CACs). Unfortunately, a number of concerns have been identified in the manuscript. The specific comments are below:

Specific comments:

1. English needs to be improved throughout the manuscript.

2.  The references 21 and 23 are the same reference.

3. "We put forward a hypothesis, that the majority of the circulating RNA was enclosed in an alternative type, i.e, Capsid-Antibody-Complexes (CACs)." - This hypotheses is based on a single report "Extracellular hepatitis B virus RNAs are heterogeneous in length and circulate as capsid-antibody complexes in addition to virions in chronic Hepatitis B patients" that the authors published earlier in JVI. There are no other reports to back up the authors claim. It should also be considered that many previous reports showed that the unenveloped HBV capsides either are not present or are very rare in the blood of individuals chronically infected with HBV. After careful examination of the above mentioned report in JVI,  it became clear that the fractionation experiments did not produce a convincing evidence, and the EM images were unclear as well (the latter actually was acknowledged by the authors). Overall, that original paper that the authors are referring to does not have a solid evidence showing that a considerable portion of serum HBV RNAs was indeed within the unenveloped capsids bound to natural anti-HBcAg antibodies. The majority of the experiments we actually conducted using the material produced by the cells replicating HBV genome and secreting the virions and unenveloped capsids. Only some experiments were performed with the serum samples from patients infected with HBV. Thus, the JVI paper failed to (i) demonstrate that majority of serum unenveloped capsids (if considerable amounts of them actually were present in sera - which was not demonstrated as well) were bound to anti-HBcAg antibodies; and (ii) isolate those antibody-bound capsids away from the virions and demonstrate that such fraction of isolated capsids actually contained HBV core antigen, and that the viral RNA molecules were incorporated inside those capsids. The experimental data obtained with the serum samples presented by the authors was unclear, inconclusive and open to alternative interpretations. Therefore, the authors actually do not have a solid proof to back up their above mentioned hypotheses as well as the conclusions made in their earlier JVI report.

4. Not all serum samples collected from the individuals chronically infected with HBV bear detectable amounts of the circulating immune complexes (CICs), but when they do - those CICs are HBV virions bound to natural anti-HBsAg antibodies. There are no reports from other labs providing the convincing evidence and clearly identifying considerable amounts of unenveloped capsids or unenveloped capsids bound to anti-HBcAg antibodies circulating in the blood of individuals chronically infected with HBV.

5. The reference to the natural "env-minus" HBV variants is irrelevant, since there are likely other variants that make HBsAg and are present in the same infected hepatocytes, or HBsAg comes from integrated HBV DNA. HBV would not be able to persist or to be transmitted in the absence of HBsAg and HBsAg-coated virions.

Reviewer 3 Report

The review article submitted by Zhang et al. is focusing on capsid-derived immune complexes accompanying hepatitis B virus (HBV) infection and their potential impact on HBV-induced liver disease. The review is exceedingly systematic, up to date and detailed. Very advantageously, predominantly in vivo data, i.e., data derived from HBV carriers and patients, are summarized. The review is very pleasant to read and takes into account not only the actual literature, but also relatively old publications in which the existence of capsid-derived immune complexes was first observed and described.

For an improvement, I have two minor suggestions:

1.         The role of the viral envelope proteins in sub/viral particle formation and release. Here, the authors state: “Our work in cell culture models suggests that the relative ratio of large, middle and small surface antigen may modulate the molecular decision on release of enveloped or unenveloped viral particles” and thereby cite their own recent publication #24. A somewhat more detailed description of the impact of the three viral envelope proteins on particle formation and their fate would benefit the article.

2.         The authors may also consider and discuss the publication from Possehl et al. (1992) which demonstrated that no naked HBV capsids were detectable in immunosuppressed patients, i.e., even in the absence of anti-HBc antibody. In addition, there are some data in the literature derived from animal models, like immunocompromised HBV-transgenic mice or HBV-infected humanized chimeric mice, demonstrating only very low levels of naked capsids (e.g.,Hong et al., 2021). 

Possehl, C., Repp, R., Heermann, K. H., Korec, E., Uy, A., & Gerlich, W. H. (1992). Absence of free core antigen in anti-HBc negative viremic hepatitis B carriers. Arch Virol Suppl, 4, 39-41. doi:10.1007/978-3-7091-5633-9_8

Hong, X., Luckenbaugh, L., Mendenhall, M., Walsh, R., Cabuang, L., Soppe, S., . . . Hu, J. (2021). Characterization of Hepatitis B Precore/Core-Related Antigens. J Virol, 95(3). doi:10.1128/jvi.01695-20

Author Response

      We thank the reviewer for his/her constructive criticism of our manuscript. As per the suggestions, we have added additional description of the impact of the L/M/S envelope proteins on the formation of virions and naked capsids. Also, we have included the two references and made relevant discussions on the discrepancies between these two and Moller’s articles. We hope the current version will be acceptable for publication.